# Model Checking Autonomous Components within Electric Power Systems Specified by Interpreted Petri Nets

**DOI:** 10.3390/s22186936

**Published:** 2022-09-14

**Authors:** Iwona Grobelna, Paweł Szcześniak

**Affiliations:** Institute of Automatic Control, Electronics and Electrical Engineering, University of Zielona Góra, 65-516 Zielona Góra, Poland

**Keywords:** control system, model checking, Petri net, power energy system, specification, verification

## Abstract

Autonomous components within electric power systems can be successfully specified by interpreted Petri nets. Such a formal specification makes it possible to check some basic properties of the models, such as determinism or deadlock freedom. In this paper, it is shown how these models can also be formally verified against some behavioral user-defined properties that relate to the safety or liveness of a designed system. The requirements are written as temporal logic formulas. The rule-based logical model is used to support the verification process. An interpreted Petri net is first written as an abstract logical model, and then automatically transformed into a verifiable model that is supplemented by appropriate properties for checking. Formal verification is then performed with the nuXmv model checker. Thanks to this the initial specification of autonomous components can be formally verified and any design errors can be identified at an early stage of system development. An electric energy storage (EES) is presented as an application system for the provision of a system service for stabilizing the power of renewable energy sources (RES) or highly variable loads. The control algorithm of EES in the form of an interpreted Petri net is then written as a rule-based logical model and transformed into a verifiable model, allowing automatic checking of user-defined requirements.

## 1. Introduction

Interpreted Petri nets, initially introduced for the specification of the control part of a cyber-physical system [1], have already proved to be useful also in the modelling of autonomous components within power energy systems [2]. These type of nets, based on the classic theory of Petri nets [3,4] and benefitting from its analysis and verification methods [5], also take into account input and output signals to communicate with the environment, other systems or their components. Input signals are then assigned to transitions (as their guards), while output signals are assigned to places. Interpreted Petri nets are safe, that is, a place may contain only one token at any time. An active place (including a token) indicates then directly the activity of output signal(s) assigned to it. This feature facilitates code generation.

Formal verification methods, on the other side, allow the discovery of any errors related to the model at an early stage of system development [6]. Such methods save time and money, as the earlier an error is detected, the lower the cost of repairing it is. The model checking technique [7] automatically verifies a model against some user-defined properties and confirms whether these are satisfied or not. If not, appropriate counterexamples are generated that allow the identification of the error source. Model checking is a well-established and appreciated method that received the prestigious Turing award (2008, Edmund Clarke, Allen Emerson, and Joseph Sifakis) [8]. It is gaining popularity also in verification of power systems, both symbolic and statistical model checking (e.g., [9,10,11]), among other more frequently used methods, such as simulations, hardware in loop (HiL), and experiments [12]. The specification of the control algorithm in a power energy system is therefore not only for its documentation, it can also be used for validation before implementation.

This work continues our previous research on the use of interpreted Petri nets, initially introduced for cyber-physical systems [1], with a deterministic modelling methodology presented in [13] which then proved to be useful for autonomous components within power energy systems [2]. In our previous paper [2] it was shown how a control algorithm for a power energy system can be specified as an interpreted Petri net and how it can then be analyzed against structural properties (focusing especially on the determinism). The case study used an autonomous system with electricity storage for the provision of a renewable energy source (RES) stabilization system service or one having highly variable loads. Herein, further results of our research are reported, showing how to apply a rule-based logical model and formally verify behavioral requirements with the model checking technique. The modelling methodology for a deterministic system specified by the interpreted Petri net proposed in [13] is extended in order to focus on formal verification and adjustment to the power energy application area.

There are also some other approaches in the power energy domain that use Petri nets (in general). Chamorro and Jimenez [14] proposed using them for load sharing control in distributed generation applications. Lopez de Alba et al. [15] applied hybrid Petri nets for contingency analysis in power systems. Zhao et al. [16] proposed a method to help decide a multi-fault rush repair robust strategy in post-disaster distribution networks using timed Petri nets with inhibitor arcs, adapted for description of the repairing process of analyzing the impact of each fault on the process. Zhumadirova et al. [17] applied colored Petri nets as formal specification of the protective device for single-phase-to-ground short circuit. Such a specification is then analyzed to check its structural properties, involving liveness, reversibility, and safeness. Beniuga et al. [18] used Petri nets for analysis of power systems protections, focusing also on structural properties, such as liveness, boundedness, or coverability. Simon et al. [19] expressed generically the networked behavior of photovoltaic systems with timed Petri nets. Another type of Petri net, namely resource-oriented Petri nets [20], were used for scheduling, e.g., for cluster tools in semiconductor manufacturing by Pan et al. [21] or for crude oil operations in refineries (hybrid, colored, timed Petri nets as an extension of resource-oriented Petri nets) by Wu et al. [22]. However, these above-mentioned approaches benefit only from Petri net formalism and available analysis methods to check structural properties of the model. They are also based on some other types of Petri nets, each one having some advantages and disadvantages. In our previous work [23], Petri nets were combined with a model checking technique for the specification of a direct matrix converter and for checking the reachability of particular states. Nevertheless, the focus was also on structural properties.

In this paper, the model checking technique is applied for verifying behavioral properties related to system safety. As far as we know, it is the first study where interpreted Petri nets and a rule-based logical model (as an intermediate format) are used to formally check behavioral properties of autonomous components within electric power systems with symbolic model checking. This allows us not only to check some structural properties of the model, but also to check the behavioral ones, regarding the relations between the various output signals or output and input signals.

The main contributions of the paper can be summarized as follows:(1)It is shown how to apply a rule-based logical model as an intermediate format between interpreted Petri net model and a verifiable model in the power system domain;(2)A novel modelling and verification methodology for control algorithms in the power energy area with the use of interpreted nets and model checking is proposed that allows verification of behavioral properties at an early stage of system development;(3)The presented idea is illustrated with a case study of an energy storage system.

The remainder of the paper is structured as follows: Section 2 presents some background on interpreted Petri nets and a rule-based logical model. Section 3 introduces a novel method for the modelling and verification of power energy system components. Section 4 illustrates the proposed approach with a case study. Section 5 presents experimental verification. Finally, Section 6 summarizes and concludes the paper.

## 2. Background

Let us introduce some definitions for easier understanding and reading as well as a simple illustrative example.

**Definition** **1** **(Petri** **net).**A Petri net [3] is a four-tuple PN = (P, T, F, M_0_), where P is a finite set of places, T is a finite set of transitions, F ⊆ (P × T) ⋃ (T × P) is a finite set of arcs, and M_0_ is an initial marking. A marking involves all places that contain a token. A transition is enabled in marking M if each of its input places contains a token. A transition can be fired if it is enabled. Then, a token is removed from all its input places and added to all its output places. A marking is reachable from any other marking if it can be reached by a sequence of transition firings.

**Definition** **2** **(liveness).**A Petri net is live [24] if from any reachable marking it is possible to fire any transition by a sequence of firings of other transitions.

**Definition** **3** **(safeness).**A Petri net is safe [24], if there is no reachable marking such that the place contains more than one token.

**Definition** **4** **(interpreted Petri net).**An interpreted Petri net [1] is a six-tuple IN = (P, T, F, M_0_, X, Y), where the first four elements describe a live and safe Petri net; X is a finite set of logic input signals, and Y is a finite set of logic output signals. A transition in an interpreted net can be fired if it is enabled and all the conditions of its input signals (assigned to it) are fulfilled.

An example of a simple interpreted Petri net is shown in Figure 1a. It can be formally written as IN_E_ = (P, T, F, M_0_, X, Y), with P = {p1, p2, p3, p4}; T = {t1, t2, t3, t4}; F = {(p1 -> t1), (t1 -> p2), (p2 -> t2), (t2 -> p3), (p3 -> t3), (t3 -> p4), (p4 -> t4), (t4 -> p1)}; M_0_ = {p1}; X = {go}; and Y = {y1, y0}.

**Definition** **5** **(strong** **determinism).**An interpreted Petri net is strongly deterministic [13] if for each reachable marking and any fixed input values, the net comes into a stable marking and at the same time there is no stable marking into which the net can come with the same input values. Additionally, for each reachable marking and any fixed input values there is only one next marking possible.

**Definition** **6** **(rule-based** **logical** **model).**A rule-based logical model [25] is a formal notation of control system behavior suitable both for formal verification (with nuXmv model checker) and prototype code generation (VHDL language for FPGA devices), consisting of the following sections: VARIABLES, INITIALLY, TRANSITIONS, OUTPUTS, and INPUTS which include:

(1)Definition of variables (places *P*, input signals *X* and output signals *Y*);(2)Initial values of variables (initial marking of the Petri net as well as initial values of the signals);(3)Rules as descriptions of transitions *T* showing how the token flow evolves;(4)Assignment of output signals to corresponding places;(5)Assignment of expected changes in input signal values to preceding places (in order to avoid the state explosion problem while model checking).

To illustrate the above definitions, a sample interpreted Petri net modelling the functionality of a two-bit counter (taken from [2]) is presented in Figure 1a with a description of signals in Figure 1b. One input signal is used to proceed with counting, and two output signals are used to control the light emitting diode lights. The corresponding rule-based logical model is shown in Figure 1c. In the rule-based logical model, the variables (lines 1–4) correspond to the set of places (*P*), input signals (*X*), and output signals (*Y*). These variables change their value according to transition firings (places, lines 9–13), activity of places (output signals, lines 14–17), or expected moment for value change (input signals, lines 18–22).

## 3. Proposed Modelling and Verification Methodology of Control Algorithms in the Power Energy Area 

The proposed modelling and verification methodology is based upon the basic modelling method presented in [2], where it is shown how to properly specify the autonomous components with power energy system with interpreted Petri nets. Here, the complete design flow from informal specification to formal verification is proposed.

The proposed methodology is shortly illustrated in Figure 2. The modelling part (blue background) was already proposed in [2] and ends with a properly formed, live, safe, reversible, and strongly deterministic interpreted Petri net. In this paper, the focus is put on the second part, namely formal verification (green background). The last stage, experimental verification (orange background), is the final verification of the implemented system and control strategy (previously formally verified) to confirm that the system operates correctly in accordance with the set algorithm in a real setup. Therefore, the general flow can be summarized as follows: information specification -> modelling -> formal verification -> experimental verification. It should be noted that formal verification is here applied before any implementation in order to make sure that the system model satisfies the user-defined requirements. If this is not the case, either necessary corrections need to be performed in the system model or the requirements need to be carefully analyzed (for an interesting case study of an incorrect requirement please refer to the catastrophe of the Lufthansa Flight LH 2904/14 in 1993 [26]).

Let us focus on the stage of formal verification, which is the main contribution of this paper. Firstly, the interpreted Petri net is written as a rule-based logical model in a text form. Places, input, and output signals are coded as variables. Then, initial marking and initial values of signals are specified. Transitions appear as rules that change the marking (activity of their input and output places). Finally, output and input signals are assigned to the appropriate places. The assignment of output signals is important from the point of view of both model checking and implementation. However, the assignment of input signals is only used in the generation of a verifiable model (in the generation of a prototype code it is omitted), in order not to face the state explosion problem. The input signals are allowed to change their value only if such a change may influence transitions firing. This aspect brings us to the problem of completeness of input signals that are guards to transitions coming out from the one particular place.

**Problem.** An interpreted Petri net model being an input for rule-based logical model may work correctly after implementation, but incorrectly during formal verification.

**Illustration.** Let us consider a simple interpreted Petri net model as shown in Figure 3 (left). Input signals x1, x2, and x3 are mutually exclusive, as they refer to the same parameter with different (disjoint) value ranges (indeed, the value of the parameter is either lower or equal or greater than the threshold). Therefore, in the real world it is not possible for any of the three input signals to be true at the same time. The interpreted Petri net then works correctly after implementation (for the sake of simplicity, what happens in places p2, p3, and p4 is omitted). 

The interpreted Petri net (Figure 3, left) is written as the rule-based logical model listed in Listing 1a with 21 lines of code. Based on that, the verifiable model in nuXmv format is generated automatically (Listing 1b), with 59 lines of code. It should be noted, that the transitions, although correct in interpretation (lines 10–12 of the rule-based logical model), result in incorrect model behavior (lines 31, 36, and 41 of the verifiable model). Therefore, when place p1 is active, then none of the places p2, p3, or p4 may be active, this structural property (satisfied in the model) can be written in CTL temporal logic as follows:AG !(p1 & (p2 | p3 | p4))(1)

Property (1) is satisfied, but it is no longer the case for any two of the places chosen from p2, p3, and p4. Checking the mutual activity of any two places from the set {p2, p3, p4} can be specified in CTL temporal logic as:AG !((p2 & p3) | (p2 & p4) | (p3 & p4))(2)

This can also be specified as results in the produced counter-example as shown in Listing 1c. Indeed, if input signals x1 and x2 are active at the same time (although not possible in the real world), both rules t1 and t2 are executed, and finally both places p2 and p3 become active, which is definitely an undesired behavior.

**Listing 1.** (**a**) Rule-based logical model of the interpreted Petri net from Figure 3 left; (**b**) generated verifiable model in nuXmv format; and (**c**) results of model checking.

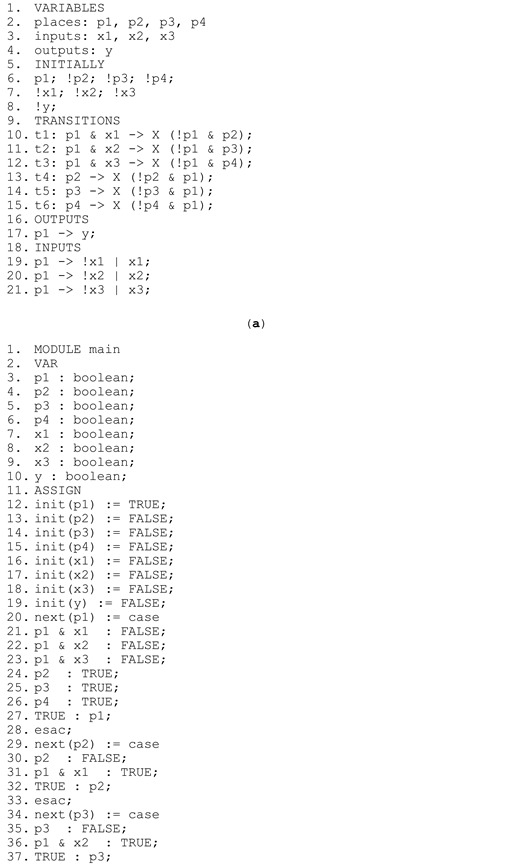



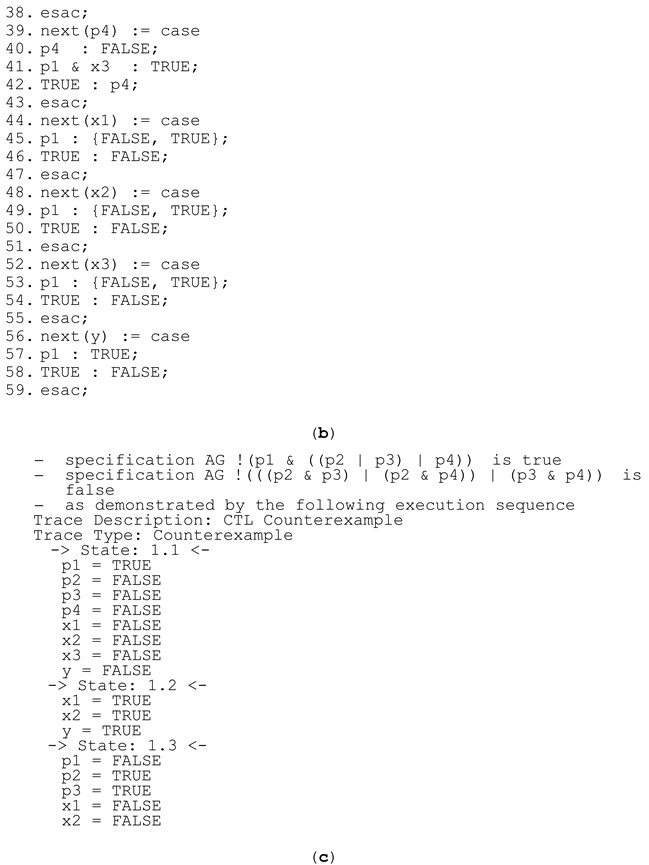



**Solution.** Although the input for preparing a rule-based logical model is a live, safe, and strongly deterministic interpreted Petri net (that works correctly after implementation), the net must be supplemented by some additional input signal assignments. Some input signals are, by their definition, disjointed, e.g., input signal x1 representing the condition “value < 1.5” and input signal x3 representing the condition “value > 1.5”. It is obvious that in the real world both signals cannot be true at the same time, as the ranges of acceptable values are disjointed, i.e., in the first case the value must be lower than 1.5, in the second one, greater than 1.5. Hence, after implementation the model would behave strongly deterministically. However, in a rule-based logical model the expected changes in input signal values are specified to simplify the model checking process. Therefore, if there are two transitions with two different input signals assigned, it is assumed in the rule-based logical model that both the input signals may change, although their interpretation is disjointed. In order to reflect this difference in perception of determinism, the interpreted Petri net needs first to be supplemented, as is shown in Figure 3 (right). In the rule-based logical model, lines 10–12 in Listing 1a are replaced with lines listed in Listing 2a. This results in changes in the verifiable model, whereby lines 20–43 of Listing 1b are replaced with lines listed in Listing 2b. Afterwards, model checking of the model reveals that both properties (1) and (2) are satisfied in the model. □

**Listing 2.** (**a**) Exchanged lines of rule-based logical model of the interpreted Petri net in Figure 3 right, with respect to Listing 1a; and (**b**) exchanged lines of generated verifiable model in nuXmv format, with respect to Listing 1b.

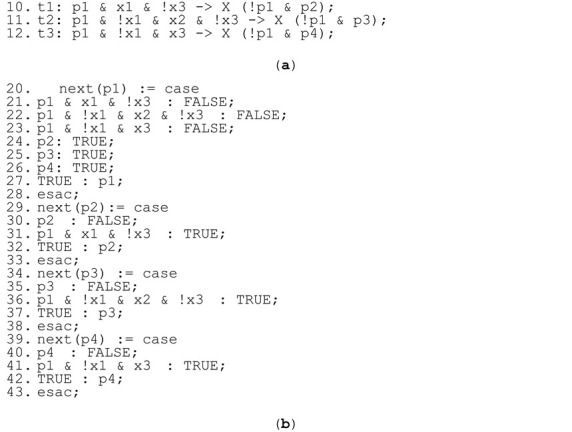



It should be noted that the rule-based logical model has a compact form and therefore it is easy to use. It is also an intermediate model that permits automatic generation of a verifiable model (and VHDL code for prototype implementation), using the implemented *m2vs* tool. The generated model is ready to be used in the nuXmv model checker [27] for simulation or formal verification. Thanks to automatic generation of the verifiable model, errors related to manually input code writing can be eliminated. The consistency between the rule-based logical model and the verifiable model is ensured through formal transformations between models (model-to-model transformation). This step is fully automatic and its output is the verifiable model.

**Theorem.** Rule-based logical model faithfully reflects the given interpreted Petri net.

**Proof.** The interpreted Petri net can be formally written as IN = (P, T, F, M_0_, X, Y) (see Definition 4). The rule-based logical model consists of five parts (see Definition 6): VARIABLES, INITIALLY, TRANSITIONS, OUTPUTS, and INPUTS. The correspondence between these two models is established as follows: 

▪Set P is directly reflected in sections: VARIABLES (definition of places) and TRANSITIONS (changing of marking), indirectly appears also in INITIALLY (with set M_0_);▪Set T is directly reflected in section TRANSITIONS (each transition is written as a separate rule that, given the conditions are satisfied, changes the marking of its input and output places);▪Set F is directly reflected in section TRANSITIONS (connection of places and transitions with each other);▪Set M_0_ is directly reflected in section INITIALLY (initial marking of all places);▪Set X is directly reflected in sections: VARIABLES (definition of input signals), INITIALLY (initial values of input signals), TRANSITIONS (as rule conditions) and INPUTS (assignment of input signals to the places where values change is expected, but it is not sure in which time point it will happen);▪Set Y is directly reflected in sections: VARIABLES (definition of output signals), INITIALLY (initial values of output signals), and OUTPUTS (assignment of output signals to the corresponding places).

From the above, it is clear that the structure of the interpreted Petri net with assigned input and output signals is consistent with the rule-based logical model. □

Safety and liveness requirements are crucial in any system with human interaction. The health and prevention of human facilities must be guaranteed to prevent any harm. Safety issues are the primary challenge that must be tackled in all systems involving humans, e.g., in robotic workstations [28,29] or spacecraft power systems [30]. A safety requirement asserts that nothing bad ever happens, and a liveness requirement, that something good will eventually happen. The requirements to be checked in the nuXmv tool are specified with temporal logic formulas [31].

So, the common desired safety property is that it should never be the case that two processes are at the same time in some critical state or that two (conflicting) output signals are active at the same time. This property (with two output signals *y1* and *y2*) can be expressed by the following LTL formula:G !(y1 & y2)(3)
or CTL formula:AG ! (y1 & y2)(4)

The common desired liveness property is that whenever a process wants to enter its critical session, it eventually does; or that whenever a request is made by the operator, the task is eventually completed by the system. This property (with input signal *request* and output signal *action*) can be expressed by the following LTL formula:G (request -> F action)(5)
or CTL formula:AG (request -> AF action)(6)

All user-defined requirements are added at the end of the generated verifiable model using the syntax of the nuXmv model checker. Finally, model checking can be performed. The system model is then verified against the requirements. If any of them is not satisfied, generated counterexamples help find the error source and correct it. Only after successful verification can the design flow proceed, and the designed system be implemented and experimentally verified.

## 4. Case Study

Let us illustrate the proposed approach with a case study of an energy storage system [32,33]. It is used to provide a system service related to the stabilization of power resulting from fluctuations in the power generated by RES or loads with large and sudden power changes. For the informal specification of the system please refer to [2].

Let us just shortly summarize the basic features: The algorithm is initiated when the permissible power fluctuations at the point of common coupling (PCC) are exceeded *ΔP_RES/LOAD_ > P_Lim_*. Correct operation of the algorithm requires the maintenance of the *SoC* of an energy storage that enables both charging and discharging. Therefore, after the algorithm is initialized, the energy level in the energy storage should be brought to 60–65% *SoC_max_*. When the required *SoC* level is achieved, the power stabilization algorithm can be started. The proposed algorithm keeps adjusting the power setpoint, which changes in an unpredictable way. As a result of the algorithm in the PCC, a constant power level *P_ref_* is maintained for periods of 1 min. The *P_ref_* value in the next minute is determined from the average RES/LOAD power that was recorded in the previous one-minute period *P_ref_*_(*n*)_ = avg(*P_RES/LOAD__*_1_*min*(*n*−1)_). The energy storage then injects energy into the grid or is charged with grid energy so as to keep the line power constant at *P_ref_*. The algorithm takes into account the maximum allowable power change *ΔP_ref_* = *P_ref_*_(*n*-1)_ − *P_ref_*_(*n*)_, which results from the short-circuit power at the PCC point and which adversely affects the flicker *P_st_*. This means that the power step change should not cause a relative voltage change in the power system of more than 3%. Therefore, the following relationship must be satisfied: *ΔP_ref_/S_kQ_ <* 0.03, *S_kQ_* is a short-circuit power at the PCC. If *∆P_ref_* ≥ *P_Lim_*, the charging/discharging power of the energy storage is limited to *P_ES_* = *P_Lim_*.

The control algorithm of power stabilization of RES and loads is written as an interpreted Petri net (shown in Figure 4, with input and output signals in Table 1) that is live, safe, reversible, and strongly deterministic, using the methods provided in [2]. The original net from [2] was here supplemented by assignment of input signals (solving the problem of the generated verifiable model presented in the previous section).

Then, the rule-based logical model was created, presented in Listing 3, with 69 lines of manually entered code. It was then automatically transformed into a verifiable model with 193 lines of code (approx. three times longer).

**Listing 3.** Rule-based logical model describing the overall supervisory energy management system of the algorithm for power stabilization of RES/LOAD based on Figure 4.

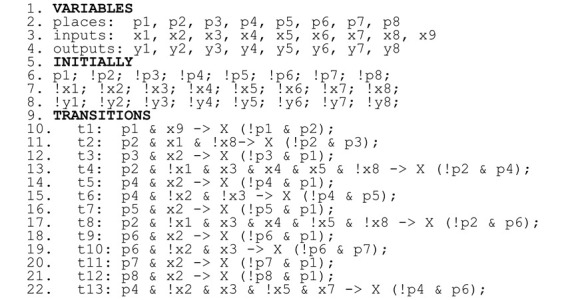



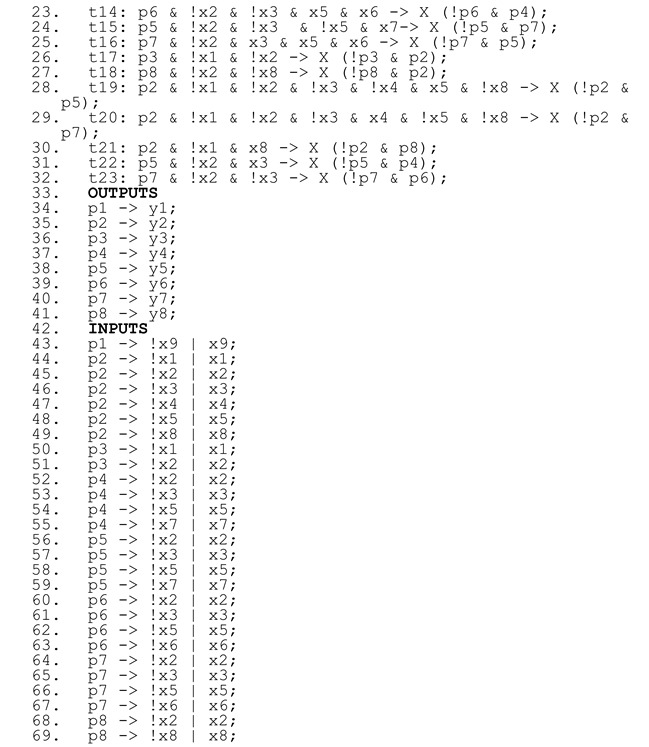



The nuXmv model can now be supplemented by user-defined requirements. Sample safety properties defined as CTL temporal logic formulas include checking whether the following situations are always true:(1)Idle state is not activated during battery pre-discharging (both output signals y1 and y8 are never active at the same time):
AG !(y1 & y8)(7)

(2)When |*P_ESmax_*| ≤ |*P_Lim_*| the battery is either charging or discharging (both output signals y4 and y6 are never active at the same time):

AG !(y4 & y6)(8)

(3)When |*P_ESmax_*| = |*P_Lim_*| the battery is either charging or discharging (both output signals y5 and y7 are never active at the same time):

AG !(y5 & y7)(9)

(4)Every request to start the algorithm (occurrence of signal x9) results eventually in initialization of initial conditions of the algorithm (output signal y2):

AG (x9 -> AF y2)(10)

(5)Every request to quit the algorithm (occurrence of signal x2) results eventually in returning to the idle state (output signal y1):

AG (x2 -> AF y1)(11)

Safety properties expressed by CTL temporal logic Formulas (7)–(9) are satisfied in the model, as well as liveness property (10). Liveness property (11) is not satisfied, as demonstrated by the generated counter example. Request to quit the algorithm does not result in going into an idle state when made in the initialization state. A weaker property, excluding initialization step, is already satisfied in the model:AG ((x2 & !p2) -> AF y1)(12)

**Listing 4.** Generated counterexample for request to quit the algorithm.

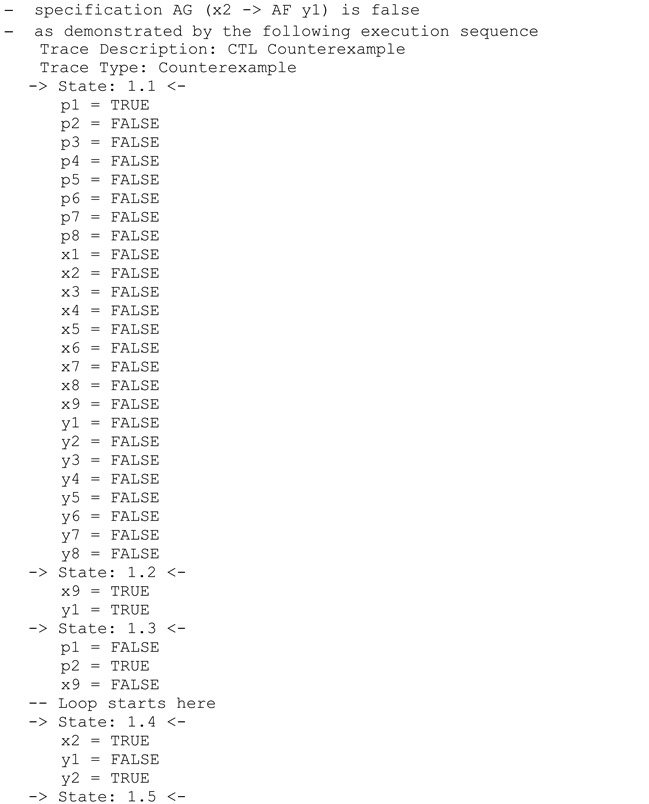



## 5. Experimental Verification

Experimental research was carried out on a prototype with batteries fabricated using lithium-titanium-oxide (LTO) [32] technology. The energy storage consists of Rechargeable Battery SCiB 23Ah cells manufactured by Toshiba. The achieved power of the tested electric energy storage was 100 kW, while its capacity was 35 kWh. Fabricated using the SiC technology, the bidirectional DC/AC power electronic converter connecting the energy storage with the power grid consists of two 50 kW inverters operating in parallel. The prototype of the energy storage was implemented on the low voltage side of the MV/LV transformer station with a rated power of 630 kVA. The experiments with the load power stabilization algorithm were carried out using the Fluke 437-II Network Parameter Analyzer.

In the test, one large load with a power of 18 kW was used as a variable load, which changed its power from zero to a nominal value within one minute. Moreover, the loads normally operating in the network were also included in the general profile of network load changes. The test results are shown in the Figure 5. Before starting the algorithm, the energy storage was set to the appropriate state of charge 0.6 *SoC_max_* ≤ *SoC* ≤ 0.65 *SoC_max_*. The physical start of the algorithm took place at the time *t*_0_, where the main contactor supplying the power electronic converter was turned on. Then, the converter and energy storage were initialized, which lasted about 4 s. During the initialization, it is not possible to interrupt the algorithm, because then the main control unit exchanges data with its peripherals and initializes them to work. Only a hard reset of the main control unit can interrupt the initialization process. At the time *t*_1_, the inverter output signals are generated, which is equal to the start of the algorithm operation.

As a result of the operation of the prototype with an energy storage and the implemented power stabilization algorithm, the power consumed from the power grid did not exceed 38 kW, with low-frequency fluctuations not exceeding 10 kW. The test results show the lack of compensation for rapid changing load power fluctuations, which results from the measuring system used in the prototype having a measurement acquisition time of 250 ms. This time is too long to successfully stabilize the rapidly changing load in the power grid.

Because the prototype of the energy storage was installed in a real power system as an autonomous, maintenance-free device, it was not possible to measure the control and set signals in the control system. Nevertheless, the implemented algorithm was verified in the testing phase using the method proposed in the article. The experimental prototype test was carried out only for load changes because the power changes generated by RES were not possible in our laboratory. The difference in the operation of both load and RES power stabilization algorithms is that the energy storage system charges when generation from RES is too high, and discharges in the case of reduced generation from RES. For the stabilization of the load power, the situation is opposite as shown in the article.

The results of the proposed approach to model checking of autonomous components within electric power systems cannot be directly compared with other approaches using any other metrics, as in the case, e.g., of short-term load forecasting, focusing on such characteristics of electric load sequence as stability and flexibility sequence [34]. Indeed, the novelty is built on application of symbolic model checking to verify components of power electronics systems. So far, the most frequently used verification methods in the industry are simulations and experiments. However, they must be performed manually and do not provide 100% confidence that the system will operate correctly in all situations. Symbolic model checking can be used to achieve a guarantee that the system satisfies user-defined requirements. It can be applied automatically before physical production of the power electronics systems, so that any errors, incorrect assumptions, or unforeseen situations are detected as early as possible. It should be emphasized that the significance of the results of symbolic model checking (in general) is correlated with the proper selection of requirements which are to be verified. The presented results of experimental tests of a real prototype operating autonomously in a low voltage network are the final verification of the implemented system and the described control strategy. The correct operation of the system signifies its verification in real conditions.

## 6. Conclusions

In the article a novel approach was proposed to formally verify behavioral properties of autonomous components within electric power systems that are specified by interpreted Petri nets (using symbolic model checking and the rule-based logical model as an intermediate format). Therefore, the previously proposed modelling methodology [2] was herein extended to the more complete modelling and verification methodology for a power system domain based on interpreted Petri nets. A rule-based logical model is applied to automatically generate a verifiable model, ready to be imported into the nuXmv model checker, in order to speed up model creation and eliminate errors related to manually entered code. The model checking technique is used to verify whether the system model satisfies user-defined behavioral requirements that are expressed as temporal logic formulas. This permits not only the checking of some basic structural properties, including reachability, but also of the behavioral properties, corresponding to the functionality of a designed system. Such properties include the cause-and-effect relationships between input and output signals and enable checking of the most desired safety situations (nothing bad will ever happen) and liveness situations (something good will eventually happen). Symbolic model checking provides us 100% confidence about the satisfaction (or lack) of the defined requirements, which is the key benefit in comparison with other traditional verification techniques applied in the power energy domain (simulations, experiments, etc.). Furthermore, it may be applied at an early stage of system development which helps to save both time and money.

The limitations of the proposed modelling and verification technique include the need for cooperation with some interdisciplinary engineers. Specific aspects of control algorithms in power energy systems should here be modelled in a way that is closer to the automatic and control domain and the engineers need to be familiar with interpreted Petri nets. Additionally, due to the nature of the rule-based logical model and the definition of expected changes in input signal values (used to avoid the state explosion problem while model checking), special attention should be paid to the proper functionality with abstract meaning of input signals, regardless of their interpretation in the real world.

Plans for the future include the continuation of interdisciplinary cooperation to introduce more advancements into the area of power energy systems. Energy storage systems are objects that are very difficult to model due to the variability of their nominal parameters, which depend on the operating conditions. These conditions affect, among others, such parameters as capacity, cyclical life, and the current charging or discharging power value. The main factors influencing the change in these parameters are cell temperature, depth of discharge (DoD), and operating currents. Therefore, further research on the use of interpreted Petri nets is needed to include the operating conditions of the energy storage. Then it will be possible to check the correct operation of not only the control algorithms, but also the operation of the Battery Management System (BMS) and its influence on the operation of the control algorithm.

## Figures and Tables

**Figure 1 sensors-22-06936-f001:**
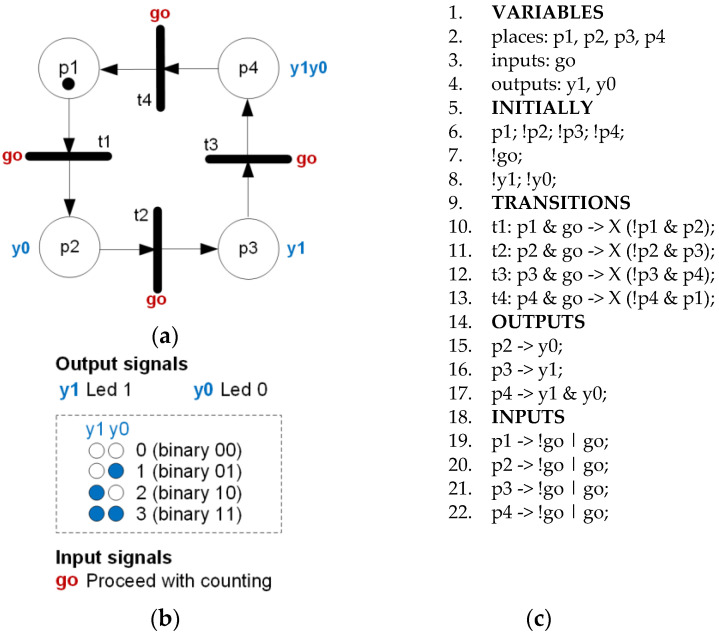
A sample interpreted Petri net (**a**), description of signals (**b**), and the rule-based logical model (**c**).

**Figure 2 sensors-22-06936-f002:**
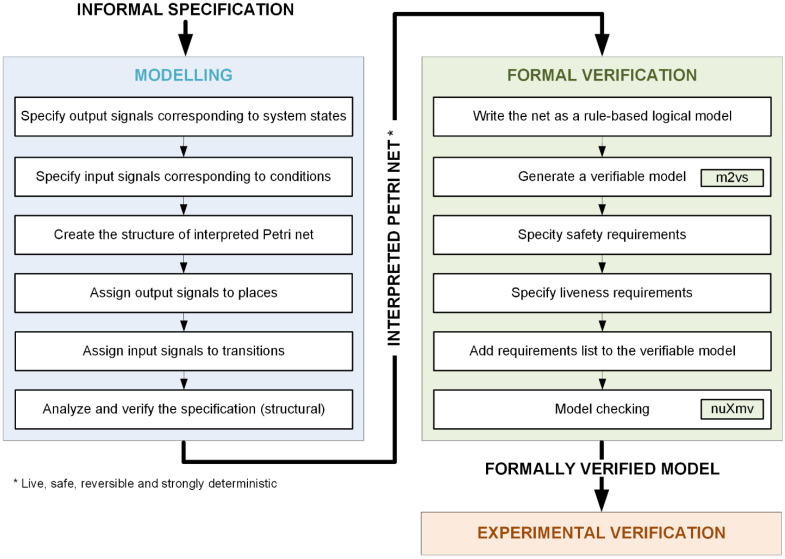
The schema of the proposed modelling and verification methodology.

**Figure 3 sensors-22-06936-f003:**
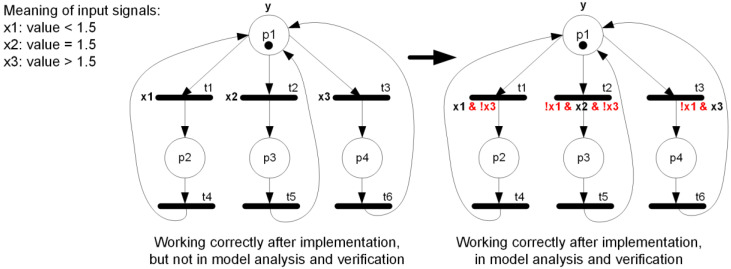
Interpreted Petri net for rule-based logical model.

**Figure 4 sensors-22-06936-f004:**
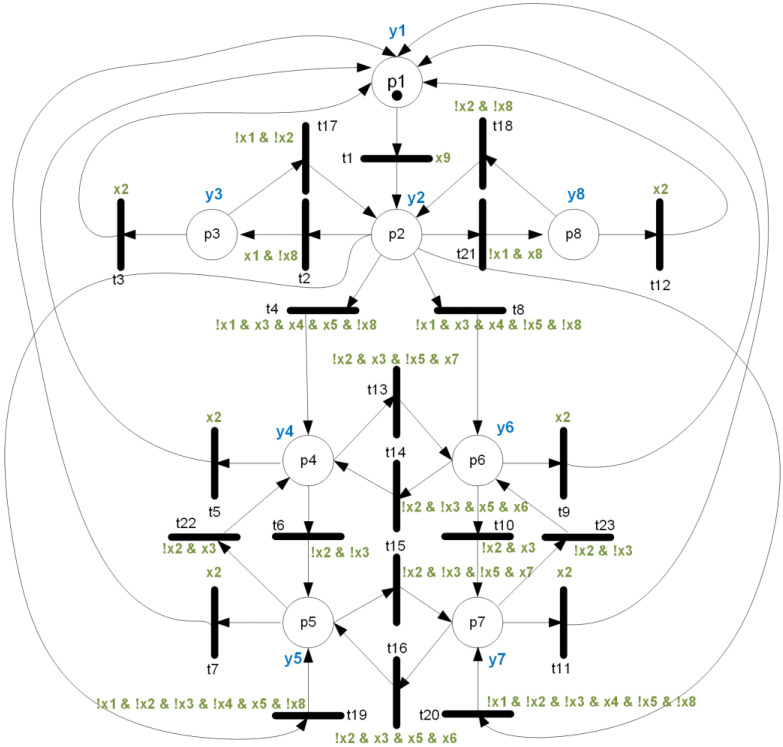
An interpreted Petri net describing the overall supervisory energy management system of the algorithm for power stabilization of RES/LOAD with all input signals assigned, prepared to be used in a rule-based logical model.

**Figure 5 sensors-22-06936-f005:**
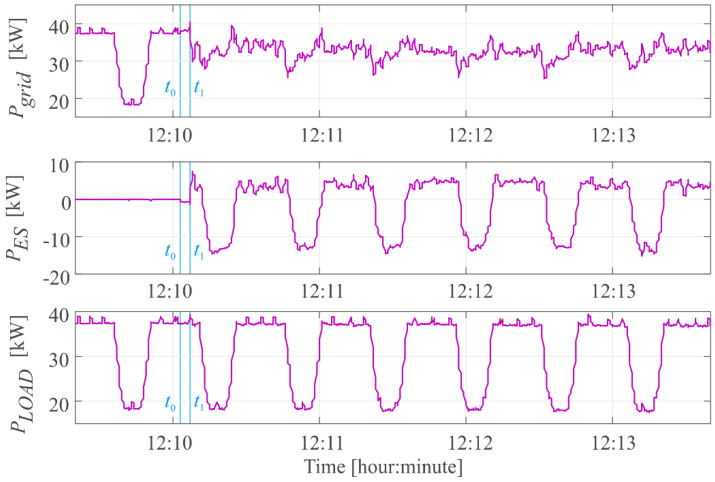
Time waveforms of the grid power *P_grid_*, power of the energy storage *P_ES_* and load power *P_LOAD_* during the operation of the power stabilization algorithm of unstable loads with energy storage prototype.

**Table 1 sensors-22-06936-t001:** Binary input and output signals.

Input	Description	Output	Description
x1	*SoC* ≤ 0.6 *SoC_max_*	y1	Idle state
x2	Request to quit the algorithm	y2	Initialization of initial conditions of the algorithm
x3	*∆P_ref_*/*S_kQ_* ≤ 0.03	y3	Battery pre-charging
x4	*∆P_RES/LOAD_* ≤ *P_Lim_*	y4	Battery charging with |*P_ESmax_*| ≤ |*P_Lim_*|
x5	*P_ES_* > 0	y5	Battery charging with |*P_ESmax_*| = |*P_Lim_*|
x6	*SoC* = *SoC_min_*	y6	Battery discharging with |*P_ESmax_*| ≤ |*P_Lim_*|
x7	*SoC* = *SoC_max_*	y7	Battery discharging with |*P_ESmax_*| =|*P_Lim_*|
x8	*SoC* ≥ 0.65 *SoC_max_*	y8	Battery pre-discharging
x9	Request to start the algorithm		

## Data Availability

Not applicable.

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
