# Peer review of "Model Checking Autonomous Components within Electric Power Systems Specified by Interpreted Petri Nets"

_sensors, 2022, doi:10.3390/s22186936_

Round 1

Reviewer 1 Report

1.       In this paper, we show how these models can also be 11 formally verified against some behavioral user-defined properties that relate to the safety or liveness 12 of a designed system., we did not use We or I in writing the scientific articles.

2.      What distinguishes interpreted Petri 34 nets from ordinary Petri nets is that they are safe, Is it correct sentence?

3.     What is the evidence of this sentence “In the article a novel approach has been proposed”?

4.     In the Conclusion Section should indicate research gaps and research directions identified as the results of research presented.

Author Response

Reviewer 1:

We would like to thank the reviewer for the valuable comments. We have carefully revised and improved the quality of the paper according to the given remarks. The changes in the manuscript are highlighted in blue. Below, we state point by point our responses to the comments, as well as the actions taken in the paper.

Comment #1: In this paper, we show how these models can also be 11 formally verified against some behavioral user-defined properties that relate to the safety or liveness 12 of a designed system., we did not use We or I in writing the scientific articles.

Response: Thank you for the comment. We have changed the phrases from the active voice into the passive voice. Additionally, the revised manuscript has been checked by a native speaker.

Action: The changes in the manuscript involve corrections related to English language (these are not explicitly marked within the text of the manuscript).

Comment #2: What distinguishes interpreted Petri 34 nets from ordinary Petri nets is that they are safe, Is it correct sentence?

Response: Thank you for the comment. In order to make the sentence clearer, we have focused on interpreted Petri nets applied in this paper and rephrased the sentence into:

Interpreted Petri nets are safe, that is, a place may contain only one token at any time.

Action: The changes in the manuscript involve lines 34-35. ‏

Comment #3: What is the evidence of this sentence “In the article a novel approach has been proposed”?

Response: Thank you for raising this issue. Indeed, there are also other approaches that use Petri nets (of different types) in the domain of power systems (as already mentioned in the article). However, we propose to use in particular interpreted Petri nets – following the notions formally introduced in 2019 for the specification of cyber-physical systems [R1] (there are also some other definitions of “interpreted Petri nets” in the literature, for example, [R2,R3]). In this manuscript, interpreted Petri nets are for the first time (as far as we know) applied to symbolic model checking of behavioral properties of autonomous components within electric power systems. The rule-based logical model is used to simplify and speed up the generation of verifiable models (this intermediate model has also been proposed in some of our previous works). This combination makes our contribution unique, allowing us to use the adjective “novel”. In order to clear up the novelty of the proposed approach, we have added the description to the conclusions as follows:

In the article a novel approach has been proposed to formally verify behavioral  properties of autonomous components within electric power systems that are specified by interpreted Petri nets (using symbolic model checking and the rule-based logical model as an intermediate format).

[R1] Grobelna, I.; WiÅ›niewski, R.; Wojnakowski, M. Specification of Cyber-Physical Systems with the Application of Interpreted Nets. In Proceedings of the IECON 2019-45th Annual Conference of the IEEE Industrial Electronics Society, Lisbon, Portugal, 14–17 October 2019; pp. 5887–5891.

[R2] Rivera-Rangel, I.; Ramírez-Treviño, A.; Aguirre-Salas, L.I.; Ruiz, J. Geometrical characterization of observability in Interpreted Petri Nets. Kybernetika 2005, 41, 553–574.

[R3] Santoyo-Sanchez, A.; Pérez-Martinez, M.A.; De Jesús-Velásquez, C.; Aguirre-Salas, L.I.; Alvarez-Ureña, M.A. Modeling methodology for NPC’s using interpreted Petri Nets and feedback control. In Proceedings of the 2010 7th International Conference on Electrical Engineering Computing Science and Automatic Control, Tuxtla Gutierrez, Mexico, 8–10 September 2010; pp. 369–374.

Action: The changes in the manuscript involve lines 512-515.

Comment #4: In the Conclusion Section should indicate research gaps and research directions identified as the results of research presented.

Response: Thank you for your suggestion. We have extended the summary part to show the research directions identified as part of the results of the presented work in the context of research on energy storage with the usage of interpreted Petri nets. Moreover, research gaps and research directions concerning the application of Petri nets in electric power systems were indicated. The following text has been added to the Conclusion section (as a separate paragraph now):

Plans for the future include the continuation of interdisciplinary cooperation to introduce more advancements into the area of power energy systems. Energy storage systems are objects that are very difficult to model due to the variability of their nominal parameters, which depend on the operating conditions. These conditions affect, among others, such parameters as capacity, cyclical life and the current charging or discharging power value. The main factors influencing the change of these parameters are cell temperature, depth of discharge (DoD) and operating currents. Therefore, further research on the use of interpreted Petri nets is needed to include the operating conditions of the energy storage. Then, it will be possible to check the correct operation of not only the control algorithms, but also the operation of the Battery Management System (BMS) and its influence on the operation of the control algorithm.

Action: The changes in the manuscript involve lines 540-550.

Reviewer 2 Report

1. Results: Recommend to be Major revisions   

This paper shows how these models can also be formally verified against some behavioral user-defined properties that relate to the safety or liveness of a designed system. The requirements are written as temporal logic formulas. The rule-based logical model is used to support the verification process. Interpreted Petri net is first written as an abstract logical model, and then automatically transformed into a verifiable model that is supplemented by appropriate properties to be checked. Formal verification is then performed with the nuXmv model checker. Thanks to that the initial specification of autonomous components can be formally verified and any design errors can be identified at an early stage of system development. An electric energy storage (EES) is presented as an application system for the provision of a system service for stabilizing the power of renewable energy sources (RES) or highly variable loads. The control algorithm of EES in form of an interpreted Petri net is then written as a rule-based logical model and transformed into a verifiable model, allowing to automatically check user-defined requirements.

This paper is with minor merits for Sensors, i.e., lacking of strong theoretical supports to clearly demonstrate the very findings to reveal its valuable contributions. It requires some major revisions.

Firstly, for Sections 1 and 2, authors should provide the comments of the cited papers after introducing each relevant work. What readers require is, by convinced literature review, to understand the clear thinking/consideration why the proposed approach can reach more convinced results. This is the very contribution from authors. In addition, authors also should provide more sufficient critical literature review to indicate the drawbacks of existed approaches, then, well define the main stream of research direction, how did those previous studies perform? Employ which methodologies? Which problem still requires to be solved? Why is the proposed approach suitable to be used to solve the critical problem? We need more convinced literature reviews to indicate clearly the state-of-the-art development.

For Section 3, authors should also introduce their proposed research framework more effective, i.e., some essential brief explanation vis-à-vis the text with a total research flowchart or framework diagram for each proposed algorithm to indicate how these employed models are working to receive the experimental results. It is difficult to understand how the proposed approaches are working.

For Sections 4 and 5, authors should use more alternative models as the benchmarking models, authors should also conduct some statistical test to ensure the superiority of the proposed approach, i.e., how could authors ensure that their results are superior to others? Meanwhile, authors also have to provide some insight discussion of the results. Authors can refer the following recommended paper,

Forecasting short-term electricity load using hybrid support vector regression with grey catastrophe and random forest modeling. Utilities Policy, 2021, 73, 101294.

Author Response

Reviewer 2:

We would like to thank the reviewer for the valuable comments. We have carefully revised and improved the quality of the paper according to the given remarks. The changes in the manuscript are highlighted in blue. Below, we state point by point our responses to the comments, as well as the actions taken in the paper.

Comment #1: This paper shows how these models can also be formally verified against some behavioral user-defined properties that relate to the safety or liveness of a designed system. The requirements are written as temporal logic formulas. The rule-based logical model is used to support the verification process. Interpreted Petri net is first written as an abstract logical model, and then automatically transformed into a verifiable model that is supplemented by appropriate properties to be checked. Formal verification is then performed with the nuXmv model checker. Thanks to that the initial specification of autonomous components can be formally verified and any design errors can be identified at an early stage of system development. An electric energy storage (EES) is presented as an application system for the provision of a system service for stabilizing the power of renewable energy sources (RES) or highly variable loads. The control algorithm of EES in form of an interpreted Petri net is then written as a rule-based logical model and transformed into a verifiable model, allowing to automatically check user-defined requirements.

This paper is with minor merits for Sensors, i.e., lacking of strong theoretical supports to clearly demonstrate the very findings to reveal its valuable contributions. It requires some major revisions.

Response: Thank you for your feedback and valuable suggestions for improvement.

Action: The changes in the manuscript involve the revisions following the next detailed comments.

Comment #2: Firstly, for Sections 1 and 2, authors should provide the comments of the cited papers after introducing each relevant work. What readers require is, by convinced literature review, to understand the clear thinking/consideration why the proposed approach can reach more convinced results. This is the very contribution from authors. In addition, authors also should provide more sufficient critical literature review to indicate the drawbacks of existed approaches, then, well define the main stream of research direction, how did those previous studies perform? Employ which methodologies? Which problem still requires to be solved? Why is the proposed approach suitable to be used to solve the critical problem? We need more convinced literature reviews to indicate clearly the state-of-the-art development.

Response: Thank you for raising this issue. To address it, we have extended the description of the existing state-of-the-art. As far as we know, our research is the first one where interpreted Petri nets and a rule-based logical model (as an intermediate format) are used to formally verify behavioral properties of autonomous components within electric power systems with symbolic model checking. This allows us not only to check some structural properties of the model (this is possible in some other approaches too), but also to check behavioral properties, regarding the relation between the various output signals or output and input signals.

To increase the quality of the manuscript, we now have a separate paragraph discussing the state-of-the-art (with added references) as well as the another one showing what novelty is introduced in this paper and which problems can be solved. The following text has been added to Section 1:

There are also some other approaches in the power energy domain that use Petri nets (in general). Chamorro and Jimenez [14] propose using them for load sharing control in distributed generation applications. Lopez de Alba et al. [15] apply hybrid Petri nets for contingency analysis in power systems. Zhao et al. [16] propose a method to help decide a multi-fault rush repair robust strategy in post-disaster distribution networks using timed Petri nets with inhibitor arcs, adapted for description of the repairing process of analyzing the impact of each fault on the process. Zhumadirova et al. [17] apply coloured Petri nets as formal specification of the protective device for single-phase-to-ground short circuit. Such a specification is then analyzed to check its structural properties, involving liveness, reversibility and safeness. Beniuga et al. [18] use Petri nets for analysis of power systems protections, focusing also on structural properties, such as liveness, boundedness or coverability. Simon et al. [19] expresses generically the networked behavior of photovoltaic systems with timed Petri nets. Another type of Petri net, namely resource-oriented Pe-tri nets [20], are used for scheduling, e.g., for cluster tools in semiconductor manufacturing by Pan et al. [21] or for crude oil operations in refineries (hybrid, colored-timed Petri nets as an extension of resource-oriented Petri nets) by Wu et al. [22]. However, these above-mentioned approaches benefit only from Petri net formalism and available analysis methods to check structural properties of the model. They are also based on some other types of Petri nets, each one having some advantages and disadvantages. In our previous work [23], Petri nets have been combined with a model checking technique for the specification of a direct matrix converter and for checking the reachability of particular states. Nevertheless, the focus was also on structural properties.

In this paper, the model checking technique is applied for verifying behavioral properties related to system safety. As far as we know, it is the first study where interpreted Pe-tri nets and a rule-based logical model (as an intermediate format) are used to formally check behavioral properties of autonomous components within electric power systems with symbolic model checking. This allows us not only to check some structural proper-ties of the model, but also to check the behavioral ones, regarding the relations between the various output signals or output and input signals.

We have also added some additional descriptions to Section 2 as follows:

An example of a simple interpreted Petri net is shown in Fig. 1(a). It can be formally written as INE = (P, T, F, M0, X, Y), with P = {p1, p2, p3, p4}; T = {t1, t2, t3, t4}; F = {(p1 -> t1), (t1 -> p2), (p2 -> t2), (t2 -> p3), (p3 -> t3), (t3 -> p4), (p4 -> t4), (t4 -> p1)};  M0 = {p1}; X = {go}; and Y = {y1, y0}.

In the rule-based logical model, the variables (lines 1-4) correspond to the set of places (P), input signals (X) and output signals (Y). These variables change their value according to transition firings (places, lines 9-13), activity of places (output signals, lines 14-17) or expected moment for value change (input signals, lines 18-22).

Action: The changes in the manuscript involve lines 64-92, 129-132 and 156-160.

Comment #3: For Section 3, authors should also introduce their proposed research framework more effective, i.e., some essential brief explanation vis-à-vis the text with a total research flowchart or framework diagram for each proposed algorithm to indicate how these employed models are working to receive the experimental results. It is difficult to understand how the proposed approaches are working.

Response: Thank you for your question. We have extended the descriptions of the proposed framework to clearly indicate each step. The following text has been added to the manuscript in Section 3:

The proposed methodology is shortly illustrated in Fig. 2. The modelling part (blue background) has already been proposed in [2] and ends with a properly formed, live, safe, reversible and strongly deterministic interpreted Petri net. In this paper, the focus is put on the second part, namely formal verification (green background). The last stage – experimental verification (orange background) is the final verification of the implemented system and control strategy (previously formally verified) to confirm that the system operates correctly in accordance with the set algorithm in a real setup. Therefore, the general flow can be summarized as follows: information specification -> modelling -> formal verification -> experimental verification. It should be noted that formal verification is here applied before any implementation, in order to make sure that the system model satisfies the user-defined requirements. If this is not the case, either necessary corrections need to be done in the system model or the requirements need to be carefully analyzed (for an interesting case study of an incorrect requirement please refer to the catastrophe of the Lufthansa Flight LH 2904/14 in 1993 [26]).

Action: The changes in the manuscript involve lines 170-183 and 188-189.

Comment #4: For Sections 4 and 5, authors should use more alternative models as the benchmarking models, authors should also conduct some statistical test to ensure the superiority of the proposed approach, i.e., how could authors ensure that their results are superior to others? Meanwhile, authors also have to provide some insight discussion of the results. Authors can refer the following recommended paper, Forecasting short-term electricity load using hybrid support vector regression with grey catastrophe and random forest modeling. Utilities Policy, 2021, 73, 101294.

Response: Thank you for raising this issue. It should be noted that formal verification based on the model checking technique only checks the correct functioning of the algorithm (to be more precise – only those properties are checked which have been defined by the user). The statistical methods are not applicable here. Any benchmarking models are also not applicable here. We use symbolic model checking to get the confirmation that some important behavioral properties (each time defined by the user and considering, e.g., the relation between output signals activity) are satisfied in the system model. Section 5 presents the results of experimental tests of a real prototype operating autonomously in a low voltage network. This is the final verification of the implemented system and the described control strategy. The system operated correctly in accordance with the set algorithm, which is illustrated in Figure 9. Inaccuracies of the system operation resulting from the long sampling time of the measured signals have been indicated in the article. Due to the inability to register all diagnostic parameters of the system, we could only present electrical signals measured at the system terminals. The manuscript presents the active powers on individual lines of the network. Therefore, we cannot add to Section 5 the results of other research methods that would compare the obtained results of experimental research. It should be emphasized that the prototype built worked correctly, which means the correct operation of the implemented control algorithm which has been already formally verified.

Additionally, considering the comparison of the results with other methods, it is not possible to directly compare our proposed approach with the other ones using any metrics, as is the case for short-term load forecasting, focusing on such characteristics of electric load sequence as stability and flexibility sequence shown in [R1]. Our novelty is built on the application of symbolic model checking to verify components of power electronics systems. So far, the most frequently used verification methods in the industry are simulations and experiments. However, they have to be performed manually and do not give 100% confidence that the system will operate correctly in all situations. The model checking technique is now gaining popularity in the area of power electronics, which has been shown in our previous paper [R2]. Symbolic model checking can be used to achieve a guarantee that the system satisfies user-defined requirements. It can be applied automatically before physical production of the power electronics systems, so that any errors, incorrect assumptions or unforeseen situations are detected as early as possible. Therefore, in our research the experimental setup was launched after formal verification of the control algorithm. At the same time, it should be emphasized that the significance of the results of symbolic model checking (in general) is correlated with the proper selection of requirements which are to be verified.

In order to reflect this comment in the manuscript, we have added some discussion to Section 5 (Experimental verification). A new paragraph has been added as follows:

The results of the proposed approach to model checking of autonomous components within electric power systems cannot be directly compared to other approaches using any other metrics, as in the case, e.g., of short-term load forecasting, focusing on such characteristics of electric load sequence as stability and flexibility sequence [34]. Indeed, the novelty is built on application of symbolic model checking to verify components of power electronics systems. So far, the most frequently used verification methods in the industry are simulations and experiments. However, they have to be performed manually and do not give 100% confidence that the system will operate correctly in all situations. Symbolic model checking can be used to achieve a guarantee that the system satisfies user-defined requirements. It can be applied automatically before physical production of the power electronics systems, so that any errors, incorrect assumptions or unforeseen situations are detected as early as possible. It should be emphasized that the significance of the results of symbolic model checking (in general) is correlated with the proper selection of requirements which are to be verified. The presented results of experimental tests of a real proto-type operating autonomously in a low voltage network are the final verification of the implemented system and the described control strategy. The correct operation of the system signifies its verification in real conditions.

[R1] 29.  Guo-Feng, F.; Meng, Y.; Song-Qiao, D.; Yi-Hsuan, Y.; Wei-Chiang, H. Forecasting short-term electricity load using hybrid support vector regression with grey catastrophe and random forest modeling, Utilities Policy 2021, 73, 101294, https://doi.org/10.1016/j.jup.2021.101294 (added reference [29] in the manuscript)

[R2] Szcześniak, P.; Grobelna, I.; Novak, M.; Nyman, U. Overview of Control Algorithm Verification Methods in Power Electronics Systems. Energies 2021, 14, 4360. https://doi.org/10.3390/en14144360

Action: The changes in the manuscript involve lines 494-510.

Round 2

Reviewer 2 Report

Authors have completely addressed all my concerns.